# Stiffness-Controlled Hydrogels for 3D Cell Culture Models

**DOI:** 10.3390/polym14245530

**Published:** 2022-12-17

**Authors:** Arto Merivaara, Elle Koivunotko, Kalle Manninen, Tuomas Kaseva, Julia Monola, Eero Salli, Raili Koivuniemi, Sauli Savolainen, Sami Valkonen, Marjo Yliperttula

**Affiliations:** 1Drug Research Program, Division of Pharmaceutical Biosciences, Faculty of Pharmacy, University of Helsinki, 00014 Helsinki, Finland; 2HUS Medical Imaging Center, Radiology, University of Helsinki and Helsinki University Hospital, 00290 Helsinki, Finland; 3Department of Physics, University of Helsinki, 00014 Helsinki, Finland; 4School of Pharmacy, University of Eastern Finland, 70210 Kuopio, Finland

**Keywords:** freeze-drying, nanofibrillated cellulose, 3D cell culture, hydrogel, biomaterials, convolutional neural network

## Abstract

Nanofibrillated cellulose (NFC) hydrogel is a versatile biomaterial suitable, for example, for three-dimensional (3D) cell spheroid culturing, drug delivery, and wound treatment. By freeze-drying NFC hydrogel, highly porous NFC structures can be manufactured. We freeze-dried NFC hydrogel and subsequently reconstituted the samples into a variety of concentrations of NFC fibers, which resulted in different stiffness of the material, i.e., different mechanical cues. After the successful freeze-drying and reconstitution, we showed that freeze-dried NFC hydrogel can be used for one-step 3D cell spheroid culturing of primary mesenchymal stem/stromal cells, prostate cancer cells (PC3), and hepatocellular carcinoma cells (HepG2). No difference was observed in the viability or morphology between the 3D cell spheroids cultured in the freeze-dried and reconstituted NFC hydrogel and fresh NFC hydrogel. Furthermore, the 3D cultured spheroids showed stable metabolic activity and nearly 100% viability. Finally, we applied a convolutional neural network (CNN)-based automatic nuclei segmentation approach to automatically segment individual cells of 3D cultured PC3 and HepG2 spheroids. These results provide an application to culture 3D cell spheroids more readily with the NFC hydrogel and a step towards automatization of 3D cell culturing and analysis.

## 1. Introduction

Drug development is an expensive and time-consuming business. Current estimations for an approved drug product are 12–15 years and $1000 million accounting for the costs of failed trials and investments [1,2]. Accurate and reliable in vitro assays are required to identify toxic or ineffective drug candidates already in the early phases of the drug development process to reduce costs, and to follow the rule of 3R (replacement, reduction, refinement) for animal testing. Furthermore, the need for fast, reliable, and efficient drug efficacy and toxicity assays, for example, primary cell-based assays, is increasing in the emerging era of personalized medicine [3]. 

Currently, a variety of in vitro assays, such as cell-based assays (cell lines, primary cells), microfluidics (immobilized-enzyme-micro-reactors, organ-on-a-chip), microsomes, and tissue extracts, are included in early drug discovery and the preclinical phase to evaluate drugs’ efficacy, toxicity, metabolism, pharmacokinetic profile, and delivery. Each method has its advantages and disadvantages [4,5,6,7]: Microfluidic systems facilitate a low volume of sample but require special equipment for the fabrication of chips and collecting the cells for analysis after experiments is difficult. Human liver microsomes provide a simple and relatively inexpensive method to study drug metabolism and they contain the metabolic enzymes of liver endoplasmic reticulum (such as cytochrome P450, Uridine 5’-diphospho-glucuronosyltransferase, and flavin monooxygenases), but they lack certain metabolizing enzymes, such as sulfotransferases. Tissue extracts provide accurate in vivo prediction of drug metabolism but they are expensive, sparsely available, and can show high biological variability. Cell cultures, for example, cancer cell lines or primary hepatocytes, provide powerful tools for drug efficacy, toxicity, and metabolism experiments. However, cell lines can show differences in the metabolizing enzymes, and the primary hepatocytes are expensive and sparsely available. Yet, developing cell culturing methods further can improve the advantages of cell cultures in drug discovery.

Traditionally, cells are used as two-dimensional (2D) tissue cultures in drug discovery. However, they represent living tissues poorly [8,9,10]. For example, cancer cells have different proliferation rates and metabolic activity, and they lack tumor tissue hypoxia in 2D cell cultures [11,12,13]. Three-dimensionally (3D) cultured cells show more in vivo-like gene transcription and morphologies than 2D cultured cells [14]. Furthermore, 3D cell cultures can provide a better prediction of drug behavior [8,9,10]. False conclusions about the drug efficacy and safety that result from using too-simple 2D cell culture models could be potentially reduced by using 3D cultured cells in the early phases of drug discovery. In addition, implementing 3D cultured cells into high-throughput screening (HTS) in early drug discovery would improve in vitro–in vivo correlation [3]. Eventually, this can reduce the costs of drug development and the need for test animals.

Subia et al. (2021) recently reviewed organ-on-a-chip applications for breast tumor research [5]. They summarized that 2D and 3D cell cultures, as well as microfluidic systems, are suitable for HTS. In another review, Brancato et al. (2020) concluded that scaffold-free, hydrogel, and microcarrier 3D cell systems are suitable for HTS while scaffold-based systems, bioreactors, and tumor-on-a-chip approaches have limited HTS options [10]. In accordance with Brancato et al., (2020), we believe that hydrogel-based 3D cell culture systems provide versatile in vitro tools suitable for HTS and implementable for drug discovery. Thus, in this research, we focus on hydrogel-based 3D cell cultures.

In general, hydrogels provide potential 3D cell culturing applications with their high content of water, porous structure, and polymer network that together facilitate the diffusion of oxygen, nutrients, and metabolic products. A wide range of natural or synthetic hydrogels has already been used for 3D cell culturing including, but not limited to, PuraMatrix™, AlgiMatrix™, HydroMatrix™, Matrigel, hyaluronic acid, collagen, gelatin, polyethylene glycol, and polyacrylamide [15,16,17,18,19,20]. One important parameter of hydrogel-based 3D cell culturing platforms is stiffness, i.e., the mechanical resistance that the cells sense. Stiffness of the tissue or artificial extracellular matrix (hydrogel) affects, for example, cell signal transduction, differentiation, morphogenesis, proliferation, and marker expression [19,21,22,23].

Certain deficiencies hinder the usability of hydrogels. First, especially animal-origin hydrogels, such as Matrigel, collagen, and gelatin, have inconsistency and batch-to-batch variation that cause problems with reproducibility [24,25]. Second, handling the hydrogels might be laborious. For example, some hydrogels require gelation with salts before they are suitable for cell seeding [15], and, if considering HTS, the viscous nature of hydrogels can be hard for automated pipetting systems [26]. Third, hydrogels that have limited transparency complicate observation, especially in high-content imaging, and require special setups for analysis [10]. Taken together, the hydrogel used in the high-throughput drug screening should be signal-free, have no human or animal origin, be easy to handle, transport, and store, and facilitate in-plate and off-plate analysis. To solve these challenges, a hydrogel that is stable at room temperature, supports cell seeding without multistep processing, is applicable for (high-content) imaging, and permits the collection of 3D cultured cell samples for off-plate analysis would be invaluable.

The platform used for 3D cell cultures is not the only aspect that hinders the implementation of cell spheroids more closely into drug discovery and development. Segmentation of cells in 3D cultures is necessary when performing, for example, drug efficacy and toxicity assays. Manual segmentation is often too overly tedious and time-consuming to be practical. In addition, although manual segmentation is often considered the gold standard, manual segmentation is subjective and the results are non-repeatable. For these reasons, various semi-automatic [27] and automatic approaches [28] for segmenting cells have been developed. However, the semi-automatic segmentation of 3D data still requires quite a lot of human work, and the results are non-reproducible. The developed automatic methods are typically problem-specific, i.e., developed for a certain type of cell. To our knowledge, there are no software or machine learning models available, at least in open-source repositories, that could automatically produce high-quality segmentations for the cells used in our studies. In addition, even when the method is suitable for a given application, clumped cells and weak imaging quality can degrade the segmentation results considerably.

We have previously shown the suitability of wood-derived nanofibrillated cellulose (NFC) hydrogel for 3D cell culturing and that the cells form spheroids in certain tissue-specific NFC concentrations [29,30]. NFC hydrogel is a shear-thinning, biocompatible, enzymatically degradable biomaterial. In addition, signaling between NFC hydrogel and cells is unspecific mechanical signaling, i.e., no specific interactions between the cells and the NFC hydrogel occur [31]. The mechanical force (stiffness) required by cells of different soft-tissue origins can be easily accessed by controlling the fiber concentration of the NFC hydrogel. Furthermore, freeze-dried NFC hydrogel forms highly porous structures, which can be stored and transported in a dry form at room temperature, and subsequently reconstituted rapidly in one step, as we have recently shown [32,33].

In this study, we aimed to freeze-dry 1.5% NFC hydrogel, reconstitute dry NFC formulations in one step with a mixture of water, cell medium, and cells into higher- or lower-fiber concentrations, and subsequently 3D culture prostate cancer (PC3), hepatocellular carcinoma (HepG2), and human adipose stem/stromal cells (hASCS) in the freeze-dried and reconstituted NFC hydrogels with the desired fiber concentration, i.e., stiffness. We aimed to show that regardless of the original NFC fiber concentration, freeze-dried and reconstituted NFC hydrogel could be used for 3D cell culturing of different cell types and tackle the challenges introduced earlier. Furthermore, to enable the segmentation of 3D cultured cells in practice, we aimed to show that the convolutional neural network (CNN)-based automatic nuclei segmentation approach, a type of supervised machine learning, is a viable option to automatically segment individual cells of PC3 and HepG2 spheroids. We hypothesized that the hydrogels obtained by reconstitution of dry NFC samples would provide a potential platform for 3D cell culturing of different cell types for applications in drug screening as a human-on-a-well-plate approach, and in the future preferably in automated format, for example in HTS.

## 2. Materials and Methods

### 2.1. Freeze-Drying and Reconstitution of NFC Hydrogel with Sucrose

Prior to freeze-drying, NFC hydrogel with 1.5% (*m*/*V*) fiber concentration (GrowDex®, UPM Kymmene, Lappeenranta, Finland) was mixed with sucrose (Sigma Aldrich, St. Louis, MO, USA) by syringe mixing, as described earlier by Paukkonen et al. (2017), to obtain the final formulation of 1.5% NFC hydrogel with 300 mM sucrose [34].

The freeze-drying cycle was programmed according to the glass transition temperature of the maximally freeze-concentrated sample (Tg’) of the NFC hydrogel formulation. Tg’ was determined with differential scanning calorimetry (TA instruments, New Castle, DE, USA) by decreasing the temperature from +40 °C to −80 °C and then heating 10 °C/min to +20 °C with a cell purge gas flow (N_2_) of 50 mL/min. Temperature and heat flow were calibrated with indium. Tg’ measurements were conducted in triplicate and results were analyzed with TRIOS software version 4.5.0.52598 (TA Instruments, New Castle, DE, USA).

The 1.5% NFC hydrogel with sucrose was freeze-dried in 6 mL ISO Clear Type I Tubular Glass Vial 2 -freeze-drying vials (Adelphi, Haywards Heath, UK) with a laboratory-scale freeze-dryer, LyoStar II (SP Scientific Inc., Warminster, PA, USA). The freeze-drying was performed by first decreasing the shelf temperature by 1 °C/min until −47 °C (freezing step). The shelf temperature was kept at −47 °C for 2 hrs. During the primary drying, the temperature was increased to −42 °C (1 °C/min) and the pressure was set to 50 mTorr (capacitance manometer). The endpoint of the primary drying was determined by comparing the pressure values of Pirani and capacitance manometer vacuum sensors. During the secondary drying, the temperature was raised 1 °C/min from −42 °C to room temperature after which the vials were closed with rubber Daikyo D Sigma freeze-drying stoppers (Adelphi, Haywards Heath, UK) in a dry nitrogen atmosphere (400,000 mTorr). The cycle lasted approximately 100 hrs. A graphical presentation of the used freeze-drying cycle can be found in the Appendix A). The closed vials were sealed manually with All-Aluminium Crimp seals (West Pharmaceutical Services, Exton, PA, USA). Freeze-dried samples were stored in closed vials, protected from light, at room temperature for up to two weeks before being processed.

Samples were reconstituted by using three different approaches depending on the desired application. (1) In the first method, freeze-dried NFC samples with sucrose were reconstituted by direct addition of milliQ (mQ)-water either gravimetrically into the NFC fiber concentration as before freeze-drying or into 3% (*m*/*V*) of fiber concentration. (2) In the second method, freeze-dried NFC samples with sucrose were reconstituted with a mixture of mQ-water and Dulbecco’s modified eagle medium (DMEM) with high glucose and L-glutamine (Gibco, Thermo Fisher Scientific, Waltham, MA, USA) into hydrogels with 0.125%, 0.4%, 0.6%, 0.8%, and 1.0% (*m*/*V*) of NFC fiber concentrations. The amount of water in the mixture was adjusted by weighing the fresh and dry NFC formulations and media was added to obtain the correct dilution. (3) The third reconstitution method was performed similarly to method 2, but the media also contained the correct concentration of cells. This method is explained in detail in the Section 2.4.describing the cell culture protocol.

### 2.2. Morphology of Freeze-Dried NFC Hydrogel

The porosity of the freeze-dried NFC hydrogel with sucrose was analyzed withscanning electron microscope (SEM). First, the freeze-dried NFC hydrogel cake was manually cut with tweezers. Then, the sample was carefully placed on a two-sided carbon tape with silver glue. Finally, the samples were sputtered with platinum for 25 s with an Agar sputter instrument (Agar Scientific 160 Ltd., Stansted, UK) and subsequently imaged in a high vacuum with 4–7 kV and 2–4 spot size with FEI Quanta 250 Field Emission Gun SEM.

### 2.3. Physicochemical and Rheological Properties of Reconstituted NFC Hydrogel Formulation

A coulometric Karl Fischer titrator (899 Coulometer, Metrohm, Herisau, Switzerland) was used to analyze the residual water content of freeze-dried samples to ensure successful drying. First, the dry samples were weighed to facilitate the calculation of the mass percent of water. Before the actual measurements, the background residual water of the methanol used to dissolve the samples was determined. For the analysis, the samples were dissolved in 1 mL of anhydrous methanol (Merck, Darmstadt, Germany), the samples were mixed vigorously, and 700 µL of the methanol was transferred to the measuring vessel.

The osmolality of the fresh and freeze-dried and reconstituted NFC hydrogel formulation with or without DMEM was measured with a manual freezing point depression osmometer (Osmomat 3000, Gonotech, Berlin, Germany). The osmometer was calibrated according to the manufacturer’s instructions with ultrapure water for 0 mOsmol/kg and 100 to 850 mOsmol/kg with calibration standards bought from the manufacturer (NaCl, Gonotech, Berlin, Germany). After calibration, 15 µL of the sample was pipetted into the measuring vessel. Triplicates were used in all measurements. The pH of the reconstituted samples was determined with pH paper by dipping the pH paper carefully into the hydrogel (Macherey-Nagel, Düren, Germany).

Storage moduli (G’) of freeze-dried and reconstituted NFC hydrogels were measured with the HAAKE Viscotester iQ Rheometer (Thermo Scientific, Karlsruhe, Germany) at a controlled temperature (25 °C). Storage modulus was determined with oscillatory frequency sweep analysis. NFC hydrogels with 3.0%, 1.5%, and 1.0% (*m*/*V*) fiber contents were measured with 2° cone geometry with a 0.1 mm gap, and NFC hydrogels with 0.8%, 0.6%, 0.4%, and 0.125% fiber contents were analyzed with double gap geometry with a 4 mm gap and a plate diameter of 25 mm. Before the measurement, constant amplitude sweeps were performed with constant angular frequency ω = 1 Hz and oscillatory stress between 1 × 10^−4^ and 500 Pa to determine the linear viscoelastic region. Frequency sweeps were performed with τ = 20 Pa (3.0% NFC hydrogel), τ = 5 Pa (1.5% and 1.0% NFC hydrogel), τ = 2 Pa (0.8% NFC hydrogel), τ =1.5 Pa (0.6% NFC hydrogel), τ = 0.8 Pa (0.4% NFC hydrogel), and τ = 0.5 Pa (0.125% NFC hydrogel). Data were processed with OriginPro software version 9.7.0.188 (OriginLab, Northampton, MA, USA).

### 2.4. Cell Culturing

PC3 (American Tissue Culture Collection, ATCC) cells were cultured in Ham’s F-12K (Kaighn’s) Medium (Gibco, Thermo Fisher Scientific, Waltham, MA, USA) supplemented with 10% (*v*/*v*) fetal bovine serum (FBSMerck, Darmstadt, Germany) and 1% (*v*/*v*) penicillin–streptomycin (P/S; Gibco, Thermo Fisher Scientific, Waltham, MA, USA). HepG2 (ATCC) cells were cultured in DMEM with high glucose and glutamine (Gibco, Thermo Fisher Scientific, Waltham, MA, USA) supplemented with 10% (*v*/*v*) FBS and 1% (*v*/*v*) P/S. Human adipose stem/stromal cells (hASC) (Lonza, Switzerland) were cultured in MEM-α medium (Gibco, Thermo Fisher Scientific, Waltham, MA, USA) supplemented with 5% of human serum (*v*/*v*) (Merck, Darmstadt, Germany). hASCs were used for the experiments in passage 5. All cells were cultured at + 37 °C and 5% CO_2_.

All three different cell types were cultured as 3D cell spheroids to study the impact of different stiffnesses of freeze-dried and reconstituted NFC hydrogels with different NFC fiber concentrations. A 1.0% (*m*/*V*) NFC hydrogel, 0.8% (*m*/*V*) NFC hydrogel, and 0.125% (*m*/*V*) NFC hydrogel were used for PC3, HepG2, and hASC cells, respectively. Cells were seeded with 100,000 cells/100 µL (PC3 and hASC) or 70,000 cells/100 µL (HepG2). Dry NFC samples with sucrose were directly reconstituted with a mixture of mQ-water, appropriate media, and cells as described above. After reconstitution, samples were homogenized by mixing with a pipette. Then, 100 µL of the mixed NFC hydrogel with cells was pipetted on low adhesion 96-well inertGrade BRAND plates^®^ (Merck, Darmstadt, Germany), and on top of the hydrogel was added 100 µL of appropriate serum-supplemented cell media. Media was changed every second day by first centrifuging the 96-well plate at 150× *g* (PC3 and hASCs) or 200× *g* (HepG2) for 6 min and then replacing the old media with fresh.

Three-dimensional control spheroids were cultured on the same plate as the sample cell spheroids. To culture the control spheroids, first, the fresh 1.5% NFC hydrogel was diluted with appropriate cell media supplemented with serum to the desired NFC fiber concentration. The control cell spheroids were cultured in the same NFC hydrogel concentrations as the sample 3D cell spheroids, i.e., 1.0%, 0.8%, and 0.125% for PC3, HepG2, and hASCs, respectively. Then, the 2D cultured cells were detached, collected, and counted, and the cell density was adjusted with appropriate cell media to the cell densities described above. The NFC and cell suspension were mixed and 100 µL of the obtained suspension was seeded on low adhesion 96-well inertGrade BRAND plates^®^ (Merck, Darmstadt, Germany). On top of the hydrogel was pipetted 100 µL of appropriate serum-supplemented cell media. The cell media was changed as described above.

For alamarBlue™ (Invitrogen, Thermo Fisher Scientific, Waltham, MA, USA) experiments, PC3, HepG2, and hASCs were cultured in 2D on a standard 96-well plate to ensure the viability of the used cells. The cells were seeded on the same day as the 3D cell spheroids with the same cell culture media as described above. The used cell densities were 10 k cells/cm^2^, 20 k cells/cm^2^, and 30 k cells/cm^2^ for PC3, HepG2, and hASCs, respectively. The media was changed on days 1, 3, 5, and 7 during the alamarBlue™ experiments.

### 2.5. Viability Assays

The viability of the 3D cell spheroids was evaluated by their mitochondrial activity and cell membrane integrity. Mitochondrial activity was studied with alamarBlue™ Cell Viability Reagent (Invitrogen, Thermo Fisher Scientific, Waltham, MA, USA). The cell media on top of the hydrogel was replaced as described above by 100 µL of alamarBlue™ solution diluted with culture media to a concentration of 20% (*v*/*v*). Cells were incubated for 4 hrs at 37 °C. After incubation, the well plates were centrifuged and 100 µL of alamarBlue™ solution was removed to plastic tubes. A 100 µL volume of cell media was added to replace the removed alamarBlue™ solution. Plastic tubes were spun for 15 s at 1500× *g* to remove any NFC fibers, after which 80 µL of alamarBlue™ solution was transferred to a black 96-well plate (Nunc^®^ MicroWell 96 optical bottom plates; Merck, Darmstadt, Germany), and fluorescence was measured using Varioskan LUX (Thermo Scientific, Waltham, MA, USA) and SkanIt RE-program 5.0 (Thermo Scientific, Waltham, MA, USA) (excitation 560 nm, emission 590 nm). The fluorescence signal was normalized to the signal from control cells and blank control samples without cells. The 2D control samples were treated similarly, but without the centrifugation step, and 10% (*v*/*v*) alamarBlue™(Invitrogen, Thermo Fisher Scientific, Waltham, MA, USA) solution diluted with cell media was used.

Cell membrane integrity was studied with a Cellstain double staining kit (Merck, Darmstadt, Germany). Cells were stained with calcein-AM (1:500) and propidium iodide (PI) (1:1000) diluted in Dulbecco’s phosphate-buffered saline without calcium or magnesium (DPBS, Gibco, Thermo Fisher Scientific, Waltham, MA, USA) by incubating them with the staining solution for 30 min at 37 °C and 5% CO2. Samples were imaged with Leica TCS SP5 II HCS-A confocal microscope (Leica Microsystems, Wetzlar, Germany) using argon 488 nm and DPSS 561 nm lasers.

### 2.6. Immunocytochemistry

Morphology and proliferation capacity of 3D cell spheroids cultured in freeze-dried and reconstituted NFC hydrogel were studied by staining their actin cytoskeleton, nuclei, and proliferation marker Ki67. The cell spheroids were cultured for 2 days (hASC) or 7 days (PC3, HepG2, and hASC) in NFC hydrogel as described above. On day 1 or 6 (one day before fixing) the NFC hydrogel was digested enzymatically with GrowDase ™ enzyme (UPM Kymmene, Lappeenranta, Finland) according to the manufacturer’s instructions. All the following incubations were performed in rotation until further notice.

The day after initiating enzymatic digestion of the NFC hydrogel, 3D cell spheroids were collected in low-adhesion plastic tubes and fixed with 4% paraformaldehyde for 20 min. After that, the cell spheroids were washed three times with 0.1% (*v*/*v*) Tween 20 (Merck, Darmstadt, Germany) in DPBS without calcium or magnesium. Cells were permeabilized with 0.1% (*v*/*v*) Triton X-100 (Merck, Darmstadt, Germany) in PBS for 8 min and then blocked for 1 h at RT with 3% bovine serum albumin (BSA; Merck, Darmstadt, Germany) and 0.3 M glycine (Sigma-Aldrich, Germany) in PBS-Tween 20 solution. After blocking, the cell spheroids were dried on a microscopy glass. The following steps were performed without rotation on a microscope glass. The samples were incubated overnight at 4 °C with anti-rabbit Ki67 (1:200; Abcam, Cambridge, UK) antibody and conjugated Phalloidin Alexa 488 (1:40; Thermo Fisher Scientific, Waltham, MA, USA) in 0.1% (*v*/*v*) Tween 20 in DPBS containing 3% (*m*/*V*) BSA.

On the following day, the cell spheroids were washed three times with 0.1% Tween 20 in PBS. After a careful washing step, the cell spheroids were incubated with Alexa Fluor 594 donkey anti-rabbit IgG (1:500; Life Technologies, Carlsbad, CA, USA) in 0.1% (*v*/*v*) Tween 20 in DPBS containing 5% (*m*/*V*) BSA for 1 h at room temperature. Followed by the incubation, the cell spheroids were washed carefully three times with 0.1% Tween 20 in PBS. After the last washing, all excess liquid was dried with tissue paper, and the cell spheroids were mounted with ProLong Diamond Antifade Mountant with DAPI (Life Technologies Carlsbad, CA, USA) and covered with cover glass (Menzel-Gläser, Braunschweig, Germany). Samples were imaged with a Leica TCS SP5 II HCS-A confocal microscope (Leica Microsystems, Wetzlar, Germany) using argon 488 nm and DPSS 561 nm lasers and UV diode. Images were analyzed and processed with ImageJ software.

### 2.7. Automatic Cell Segmentation

A novel CNN-based automatic nuclei segmentation approach was applied to three PC3 and HepG2 spheroids [35]. The approach used CNN models to create binary nuclei masks and seeds and utilized a marker-controlled watershed algorithm [36] to produce instance segmentation of the nuclei. The approach was trained and validated with data including twelve HepG2 cell spheroids introduced by Kaseva et al. (2022) [35]. The nuclei and imaging configurations of these spheroids were similar to the ones in HepG2 and PC3 spheroids, making the approach applicable to this study.

The spheroid images were preprocessed by first resizing the voxels near isotropic. To achieve this, the x–y-planes of both spheroids were resized to 512 × 512 voxels and the z-plane of the HepG2 spheroid was expanded by a factor of two simply by copying each x–y-plane. This procedure was not exactly the same as discussed in [35], but sufficient for a visual demonstration. Then, a 256 × 256 × N sized region of interest (ROI) was chosen from the center of each image. The height N was chosen manually to exclude x–y-planes with no visible nuclei. The segmentations of ROIs themselves were produced automatically. The computation time for each segmentation was approximately half a minute. The computed results were compared visually to the original confocal microscope images.

### 2.8. Statistical Analysis

Statistical significance was determined with Microsoft® Excel® software version 2211 (Build 16.0.15831.20098) (Microsoft, Redmond, WA, USA) with a two-tailed independent samples *t*-test where *p* < 0.05 was considered significant.

## 3. Results

### 3.1. Freeze-Dried NFC Hydrogel Formed a Porous Scaffold with Suitable Physicochemical Properties for 3D Cell Culturing after Reconstitution

Freeze-dried 1.5% NFC hydrogel with 300 mM sucrose formed a highly porous and regular scaffold (Figure 1A). The appearance of the freeze-drying cakes was elegant (Appendix A). The pH and osmolality of the freeze-dried and reconstituted NFC hydrogels were evaluated to observe the suitability of the NFC hydrogels for 3D cell culturing. The volume of media was adjusted to obtain osmolality values of 300 mOsmol/kg. The osmolality of the initial formulation (1.5% NFC hydrogel with 300 mM sucrose) was 346 mOsmol/kg (342–350 mOsmol/kg). The osmolalities and pH values of fresh and freeze-dried and reconstituted NFC hydrogels are reported in Table 1.

Storage moduli (G’) of the fresh and freeze-dried and reconstituted NFC hydrogels were measured to evaluate the successful reconstitution of NFC hydrogel into different fiber concentrations. Importantly, the storage moduli were higher than the loss moduli in all freeze-dried and reconstituted samples indicating the successfully recovered viscoelastic characteristics (Appendix A). Interestingly, NFC hydrogels with sucrose, DMEM, and water had relatively higher storage moduli when compared with NFC hydrogels with sucrose and water. The storage moduli of freeze-dried and reconstituted NFC hydrogels were slightly lower than those of corresponding fresh NFC hydrogels (Figure 1B). However, when storage modulus was correlated with the fiber concentrations, we observed that the behavior of the freeze-dried and reconstituted NFC hydrogels followed the same pattern as the fresh NFC hydrogels (Figure 1B). Figure 1D shows that the storage moduli of the NFC hydrogels increased when the NFC hydrogel concentration increased. By plotting the storage moduli of NFC hydrogel with sucrose and DMEM at the angular velocity of 4.93 rad/s with different NFC concentrations on semilogarithmic coordination we obtained Equation (1):(1)logG'=a+bc(NFC)
in which a and b are experimentally determined values and c(NFC) is the concentration of NFC hydrogel as *m*/*V*-%. For the fresh NFC with DMEM (supplemented with 10% FBS and 1% P/S antibiotics) *a* is 0.599 (Standard error (SE) ± 0.173) Pa and *b* is 2.07 (SE ± 0.26) m^2^/s^2^ (R^2^ = 0.954). For the freeze-dried and reconstituted NFC hydrogel samples with DMEM (supplemented with 10% FBS and 1% P/S antibiotics) the value of *a* is 0.298 (SE ± 0.192) Pa and *b* is 1.85 (SE ± 0.29) m^2^/s^2^ (R^2^ = 0.931). The correlation between the fiber concentration and measured storage modulus remained nearly unchanged after freeze-drying and reconstitution.

### 3.2. Cell Spheroids Cultured in NFC Maintained their Viability at Least for 7 Days and Formed Cell-Type Typical Spheroids

PC3, HepG2, and hASC cells were cultured in 1.0%, 0.8%, and 0.125% NFC hydrogels, respectively, and in both fresh and freeze-dried and reconstituted NFC hydrogels to study any differences in the viability of the spheroids. Metabolic activity of the PC3, HepG2, and hASC cells cultured in freeze-dried and reconstituted NFC hydrogel was 102%, 104%, and 88%, respectively, on day 7, when normalized to the metabolic activity of the control cells on the same day (Figure 2). When normalized to the metabolic activity of day 1 cells of the same culture, i.e., sample normalized to sample day 1, and control normalized to control day 1, metabolic activities of 127% (control) and 156% (sample) were obtained for PC3, 145% (control) and 181% (sample) for HepG2, and 255% (control) and 206% (sample) for hASCs on day 7. This indicates that for the cancer cell lines PC3 and HepG2, the metabolic activity of the cells increased relatively more in the freeze-dried and reconstituted samples than in the control.

Statistically significant differences were observed between the 3D cell spheroids cultured in fresh and freeze-dried and reconstituted NFC on days 3 and 5, and on days 3, 5, and 7, for PC3 and hASC spheroids, respectively. However, when evaluated with live/dead staining using confocal microscopy, both cell lines showed equal viability and individual spheroids had approximately 100% viability in all cell lines (Figure 2). The 3D cultured PC3 cells grew with loose morphology, HepG2 spheroids with tight morphology, and hASC spheroids with tight and round morphology.

Immunocytochemistry staining was performed to obtain sophisticated images of the morphology and proliferation capacity of 3D cell spheroids cultured in hydrogels with different stiffnesses and to compare morphology between 3D cell spheroids cultured in fresh and freeze-dried and reconstituted NFC hydrogel.

PC3 cells grew with grape-like morphology in 1% NFC hydrogel. Grape-like morphology was observed using actin and nuclei staining (Figure 3) [37]. Actin cytoskeletons of individual cells could be recognized, and the cells were not tightly bound. In addition, cell nuclei were disorganized. No difference was observed in the morphology or number of proliferating PC3 cells cultured in fresh or freeze-dried and reconstituted NFC hydrogel.

HepG2 cell spheroids grew with mass spheroid morphology in the 0.8% NFC hydrogel (Figure 4). Mass-like morphology could be observed by a tighter actin cytoskeleton network throughout the cell spheroid and with the higher organization of the cells observed by the location of the nuclei [37]. However, some disorganization was still observed, which indicated the mass-like morphology. Again, no difference was observed between HepG2 cell spheroids cultured in the fresh or freeze-dried and reconstituted NFC hydrogel.

The hASC spheroids grew with an organized structure indicating round morphology [37] in 0.125% NFC hydrogel (Figure 5). Already on day 2, hASCs had formed 3D cell spheroids. Polarization of hASCs on the perimeter of the 3D cell spheroid and their adjustment as spheroids could be observed already on day 2 from the appearance of the actin cytoskeleton and nuclei. The actin cytoskeleton formed round structures around the 3D spheroid. On day 7 the spheroids were extensive in terms of their diameter. The morphology of the 3D cell spheroids was round also on day 7 despite some loosely attached cells observed on the perimeter of the 3D cell, i.e., the structure of hASC spheroids appeared tighter on day 2 than on day 7. In addition, the number of spheroids was lower on day 7, and the size of individual cell spheroids was larger. Proliferating cells were observed only sparsely, on both days 2 and 7. In general, more proliferating cells were observed in cancer cell spheroids than in primary adipose stem cell spheroids.

### 3.3. Automatic Segmentation of Different Cell Types

The quality of the segmentation of 3D PC3 and HepG2 cell spheroids was visually assessed (Figure 6). The ground truths for PC3 spheroids were not available nor created and quantitative evaluation scores could not be computed. However, the qualitative visual assessment of the segmentation results was satisfactory. The segmentation itself was fully automatic and non-subjective. Visual evaluation was performed to verify that the results of the automatic segmentation were acceptable. While some segmentation mistakes were present, most of the segmented cells were visually reasonable. The results emphasized that the automatic segmentation of 3D cell cultures discussed in this work can be viable. In order to exploit this approach on hASC spheroids as well, the training and evaluation set of the approach should be extended to include similar nuclei as in these spheroids. Automatic segmentation indicated clearly the different morphologies of the 3D cell spheroids. The grape-like morphology and looser nuclei structures of PC3 spheroids and the mass spheroid morphology and denser packing of the nuclei of HepG2 spheroids were easily observed from the automatic segmentation.

## 4. Discussion

Most drugs that fail in the clinical phase lack efficacy or their pharmacokinetic profile is unsuitable. More precise screening of these properties in early drug discovery would reduce the need for animal testing and the costs of drug development. Novel approaches to reduce the gap between in vitro and in vivo experiments in drug discovery and research are required to address these issues. In this study, we showed the suitability of freeze-dried wood-derived nanofibrillated cellulose hydrogel for 3D cell culturing with a rapid one-step cell-seeding process. In addition, we demonstrated the possibility to reconstitute NFC hydrogel into different fiber concentrations and the possibility to estimate the stiffness of the hydrogel with a certain fiber concentration. Lastly, we showed the suitability of CNN-based models to automatically segment individual cells of multicellular HepG2 and PC3 spheroids.

Stiffness and porosity are recognized as important parameters of the biomaterials used for cell culturing. The stiffness of the material affects proliferation, growth, cell migration, differentiation, and organoid formation [23,38]. The storage moduli (describing the stiffness) of the NFC hydrogel samples with the cell media were relatively higher than the storage moduli of the samples with only water. This was observed with both fresh and freeze-dried and reconstituted samples. Interestingly, the opposite has been observed in crosslinked biomaterials. Hastruk et al. (2020) observed that higher ionic strength resulted in lower storage moduli [39]. However, NFC hydrogel is not cross-linked, which can explain the observed difference. The relatively higher storage moduli in our study can result from interactions between the fibers when the media is present. The slight differences observed with the fresh and freeze-dried and reconstituted hydrogels with the same fiber concentration could result from the orientation of the NFC polymers after freeze-drying and reconstitution. A lesser decrease in the stiffness of the freeze-dried hydrogel is observed if the polymers have an anisotropically extruded orientation [40]. In our study the freezing was isotropic, and the NFC polymers were randomly aligned after freeze-drying and especially after reconstitution. This can explain the observed decrease in the storage moduli of the hydrogels. Furthermore, minor inaccuracy in the weighing of the sample and subsequent reconstitution also affects the observed stiffness. Nevertheless, despite the differences in the storage moduli, no difference was observed in the viability or morphology of the 3D cell spheroids cultured in the fresh NFC hydrogel and in the freeze-dried and reconstituted NFC hydrogel.

As stated above, porosity is another important material attribute when considering freeze-dried biomaterial scaffolds [29]. Wu et al. (2010) showed in their study how the porosity of the scaffolds could be controlled by changing the gelatin concentration and crosslinking before unidirectional freeze-drying [41]. In addition, by changing the process parameters of freeze-drying, especially by changing the freezing step, different porosities of the material can be obtained [42]. We have previously characterized the porosity of the freeze-dried NFC formulations throughout and similar results were observed in this study evaluated from the SEM micrographs [32,33,43]. The similar porosities observed in the SEM micrographs of this study and our previous studies underline the reproducibility of the freeze-drying cycle and sample preparation. Porosity directly affects the cellularization of the cell scaffolds, for example, cell attachment and viability [44,45]. However, our results indicate that the pore size is unspecific for cell types, yet the porosity can be considered vital to facilitate mass transport in the scaffold. We believe that in our study the pore size played a less significant role than in the other studies of dry scaffolds because we cultured the 3D cell spheroids eventually in the hydrogel despite starting from the dry NFC scaffold. Parisi et al. (2021) discussed in their recent review article how different material manufacturing methods and pore size affect cellularization by cell colonization or encapsulation [46]. They concluded that the cellularization is impaired below a certain threshold of porosity and cytocompatibility of the used method. In this study, we showed that the porosity of the freeze-dried NFC hydrogel was suitable for all three different types of cells despite the diverse tissue origins of the used cells.

The metabolic activity of 3D cultured cells was lower than 2D cultured cells and the number of proliferating cells in the 3D cell spheroids was low. Yet, the viability of the cells in the spheroid observed from the live/dead staining was nearly 100%. We think that this indicates the closer relevance of 3D cell tumors to in vivo than unlimitedly proliferating 2D cell cultures [47]. For example, if the studied cells have artificially too high metabolic activity, drug candidates affecting them may show falsely positive effects resulting later in a failure in clinical trials. The statistical significance observed in the metabolic activities can be also explained by the different cell densities used in the 2D and 3D cell cultures. Subia et al. (2021) concluded in their review considering breast tumor-on-a-chip applications that tumor-on-chip models can compensate for 2D and 3D models [5]. Combining the fluidic system into the well plate with 3D culture spheroids would be a viable option to obtain an even more relevant human-on-a-plate approach suitable for HTS. Furthermore, combining the automatic segmentation with the HTS screening would hasten the analysis of HTS results.

In this study, segmentation results on one HepG2 spheroid and one PC3 spheroid were illustrated. The results were produced by manually choosing representative 3D ROIs from both spheroids and applying the 3D nuclei segmentation system introduced by Kaseva et al. [35] to these ROIs. The system operates automatically. At this stage, no ground truths have been generated for the spheroids and visual inspections of the results were performed. The inspections revealed that the system could, despite being trained with a small-sized dataset, obtain reasonable results. This is a promising outcome since it outlined that the automatic segmentation of HepG2 and PC3 spheroids could be viable in the future.

To summarize, according to our results, freeze-drying of NFC hydrogel results in suitable porosity for cell colonization and facilitates subsequent reconstitution and adjustment of the stiffness of the hydrogel to promote 3D cell culturing of different cancer cell lines and primary cells. In addition, we showed the suitability of the material for microscopy imaging and subsequent CNN-based automatic cell segmentation. These properties are important for the materials used in in vitro experiments for drug discovery in the near future. Material sciences promote drug delivery applications, for example, with vaccines and contraception [48]. We believe that in addition to the applications reviewed by Sadeghi et al. (2021), material sciences will become increasingly important in drug discovery [48].

## 5. Conclusions

Freeze-dried NFC hydrogel can be reconstituted to different stiffnesses relative to the fiber concentration. The freeze-dried and reconstituted NFC hydrogel can be used for 3D cell culturing of different human-origin cell types, such as primary hASCs, PC3 cells, and HepG2 cells, of which PC3 and HepG2 cell spheroids were suitable for automatic cell segmentation. The porosity of the material was sufficient to facilitate mass transfer and colonization of the material by the seeded cells. The 3D cultured cell spheroids showed cell-typical morphology, and first increasing and then stable metabolic activity with nearly 100% viability. Taken together, freeze-dried NFC hydrogel offers a versatile 3D cell culturing platform for cells of different origins.

## Figures and Tables

**Figure 1 polymers-14-05530-f001:**
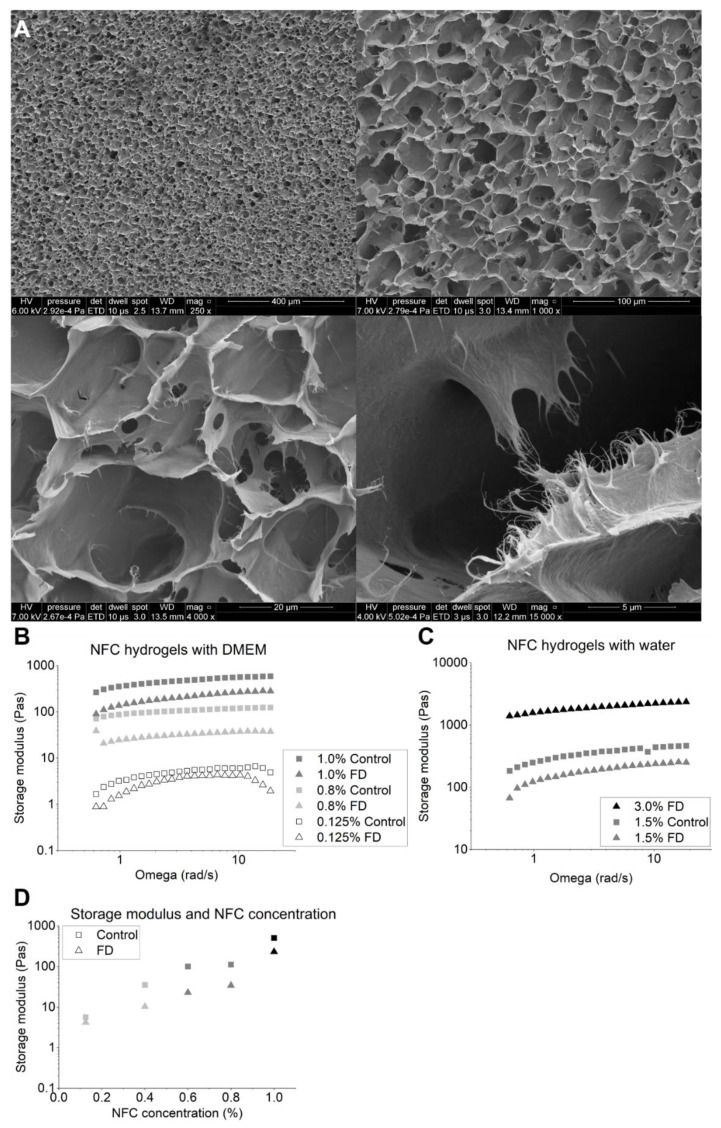
(**A**) Scanning electron microscope (SEM) micrographs of freeze-dried 1.5% nanofibrillated cellulose (NFC) samples with sucrose demonstrating a continuous porous structure of the freeze-dried sample. (**B**) Storage moduli of NFC hydrogels with sucrose and Dulbecco’s modified eagle medium (DMEM). Standard deviations (SD) of the measurements varied between 264.5–588.4 (1.0% control), 65.4–115.1 (1.0% FD), 71.6–124.2 (0.8% control), 16.5–44.6 (0.8% FD), 1.7–6.7 (0.125% Control), and 0.5–4.5 (0.125% FD) (**C**) NFC hydrogels with sucrose only. SD of the measurements varied between 256.8–462.8 (3.0% FD), 183.4–466.0 (1.5% Control), and 38.2–68.2 (1.5% FD). (**D**) Storage moduli of NFC hydrogel with sucrose and DMEM at the angular velocity of 4.93 rad/s (fresh and freeze-dried and reconstituted) with different concentrations. FD: Freeze-dried. Control: Fresh NFC formulation diluted from 1.5% NFC with 300 mM sucrose.

**Figure 2 polymers-14-05530-f002:**
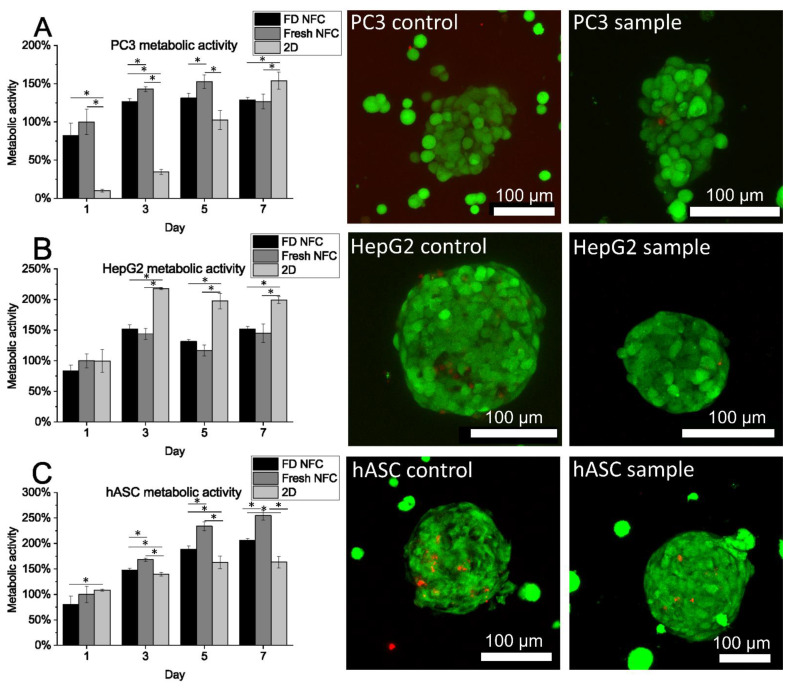
(**A**) Metabolic activity and viability of prostate cancer (PC3) cells cultured in freeze-dried and reconstituted 1.0% nanofibrillated cellulose (NFC) hydrogel, fresh 1.0% NFC hydrogel, and in 2D format. (**B**) Metabolic activity and viability of hepatocellular carcinoma (HepG2) cells cultured in freeze-dried and reconstituted 0.8% NFC hydrogel, fresh 0.8% NFC hydrogel, and 2D format. (**C**) Metabolic activity and viability of human adipose stromal/stem cells (hASCs) cultured in freeze-dried and reconstituted 0.125% NFC hydrogel, fresh 0.125% NFC hydrogel, and 2D format. Live/dead staining was performed on day 7 for all cell lines. Green: viable cells. Red: Dead cells. FD: Freeze-dried. Columns represent average values normalized to the metabolic activity of the specimen of day 1 fresh NFC. Error bars: standard deviation. Asterix: Statistically significant difference (two-tailed independent samples t-test, *p* < 0.05 considered significant).

**Figure 3 polymers-14-05530-f003:**
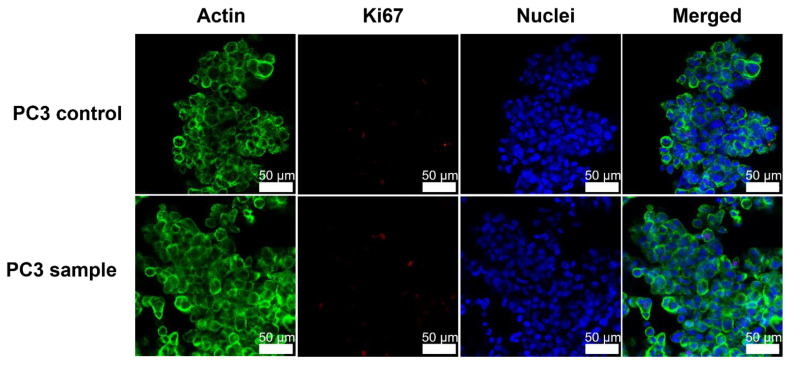
Immunocytochemical staining showing loose actin (green), proliferating cells (red), and nuclei (blue) of the 3D cultured prostate cancer (PC3) cell spheroids cultured either in 1.0% NFC hydrogel (control) or in freeze-dried and reconstituted 1.0% NFC hydrogel (sample). Green: Phalloidin conjugated Alexa Fluor 488. Red: Alexa Fluor 594. Blue: DAPI.

**Figure 4 polymers-14-05530-f004:**
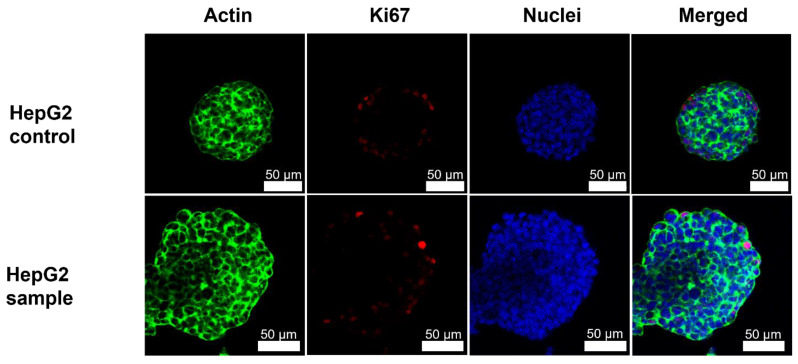
Immunocytochemical staining shows robust cell adhesion by actin (green), some proliferating cells (red) on the outer sphere of the cell spheroid, and disorganized nuclei (blue) locations of HepG2 cell spheroids cultured in fresh 0.8% NFC hydrogel (control) and freeze-dried and reconstituted 0.8% NFC hydrogel (sample). Green: Phalloidin conjugated Alexa Fluor 488. Red: Alexa Fluor 594. Blue: DAPI.

**Figure 5 polymers-14-05530-f005:**
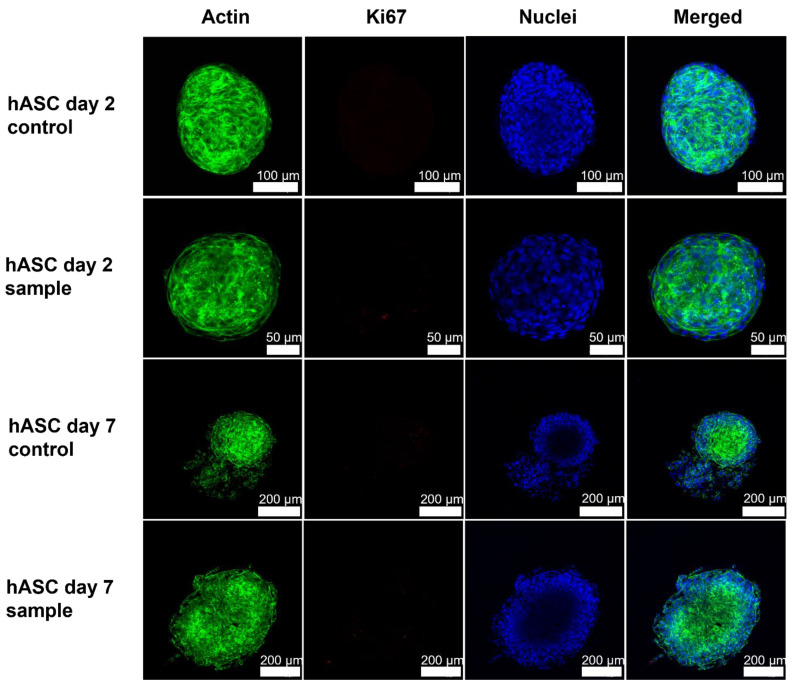
Immunocytochemical staining showing an organized structure of actin (green) and polarization of the cell observed as the shape of the nuclei (blue) of human adipose stromal/stem cells (hASCs) in fresh 0.125% NFC hydrogel (control) and freeze-dried and reconstituted 0.125% NFC hydrogel (sample). Proliferating cells (red) were observed only sparsely on both days 2 and 7. Green: Phalloidin conjugated Alexa Fluor 488. Red: Alexa Fluor 594. Blue: DAPI.

**Figure 6 polymers-14-05530-f006:**
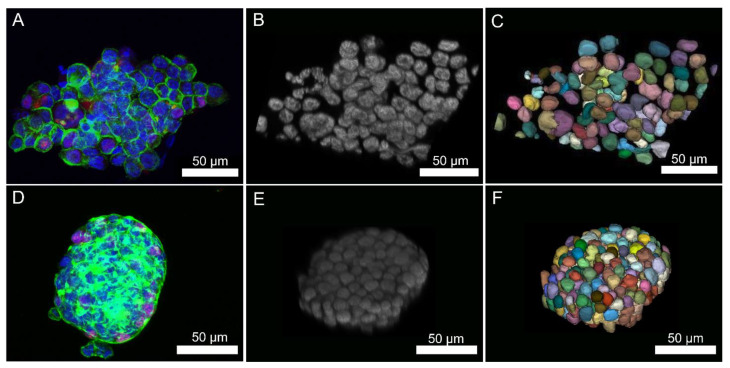
(**A**) PC3 cell spheroid cultured in 1.0% nanofibrillated cellulose (NFC) hydrogel for 7 days. Nuclei were stained with DAPI (blue), Ki67 proliferation marker with Alexa Fluor 594 (red), and actin cytoskeleton with Phalloidin conjugated Alexa Fluor 488 (green). (**B**) A grayscale figure of nuclei of PC3 spheroid for automatic segmentation. (**C**) Automatically segmented nuclei of the multicellular 3D PC3 cell spheroid. (**D**) HepG2 cell spheroid cultured in 0.8% NFC hydrogel for 7 days. Nuclei were stained with DAPI (blue), Ki67 proliferation marker with Alexa Fluor 594 (red), and actin cytoskeleton with Phalloidin conjugated Alexa Fluor 488 (green). (**E**) A grayscale figure of nuclei of HepG2 cell spheroid for automatic segmentation. (**F**) Automatically segmented nuclei of the multicellular 3D HepG2 cell spheroid.

**Table 1 polymers-14-05530-t001:** The osmolalities in the form of average (range) and pH of fresh and freeze-dried and reconstituted nanofibrillated cellulose (NFC) hydrogels. FD: Freeze-dried. DMEM: Dulbecco’s modified eagle medium.

Formulation	Osmolality (mOsmol/kg) (*n* = 3)	pH
Fresh 1.5% NFC, sucrose	346 (342–350)	7
Fresh 1.0% NFC, DMEM, sucrose	352 (368–379)	7
Fresh 0.8% NFC, DMEM, sucrose	352 (344–365)	7
Fresh 0.125% NFC, DMEM, sucrose	343 (341–345)	7
FD and reconstituted 1.5% NFC, sucrose	315 (302–338)	7
FD and reconstituted 3.0% NFC, sucrose	562 (538–586)	7
FD and reconstituted 1.0% NFC, DMEM, sucrose	314 (292–328)	7
FD and reconstituted 0.8% NFC, DMEM, sucrose	328 (319–335)	7
FD and reconstituted 0.125% NFC, DMEM, sucrose	340 (337–345)	7

## Data Availability

Not applicable.

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
