# Peer review of "Stiffness-Controlled Hydrogels for 3D Cell Culture Models"

_polymers, 2022, doi:10.3390/polym14245530_

Round 1

Reviewer 1 Report

An excellent manuscript overall.  A few comments / edits / suggestions:

Line 101: replace cells with cell.

Line 104: replace data with application?

Line 147: was the sample temperature -47C or was this the shelf temperature setpoint?

Line 148: Was the pressure controlled by the CM gauge or the Pirani?

Line 215: replace 'with' by 'at'

Line 221: replace 'with' by 'at'

Reviewer 2 Report

The manuscript by Merivaara et. al reports the results of evaluating different formulations of freeze-dried and reconstituted nanofibrillated cellulose (NFC). Although the authors conducted several experiments, it is strongly recommended to include the following observations and experiments.  

1. In the introduction (lines 47-50), the authors mentioned the limitations of current available in vitro cell-based assays for drug testing. A more detailed discussion should be included.

2. Please, in the introduction (line 57), explain in more detail the advantages of using 3D models

3. It is recommended to include a diagram or figure to show the different conditions of the study, which are explained in the section Freeze-drying and reconstitution of NFC hydrogel with sucrose. Also, in the results, include a picture showing the morphology of the hydrogels before and after the freeze-dried processing. 

4. Please, replace the term chapter 2.4 by section 2.4 (line 168). Also, this numeration is not found 

5. It is not clear how the authors measured the osmolarity and pH of the freeze-dried and reconstituted NFC hydrogels. 

6. It is recommended to explain in detail the fabrication of the control spheroids. 

7. It is recommended to include the data obtained for osmolarity and pH in a table. 

8. Please, report the used cell density in cells per cm2 

9. The authors mentioned that each NFC hydrogel formulation was evaluated by employing a different cell type. Please, explain the reasoning for this. 

10. In the results, the authors include the data obtained for the rheological studies. However, it is not explained how the a and b values were determined for each NFC hydrogel formulation. Also, it is recommended to explain in more detail figures 1B, 1C and 1D as well as including the standard deviations or significant differences in this figure. 

11. Viability of the spheroids are explained as percentages in the text, but shown as normalized ratios in Figure 2. It is recommended to use the same format when showing and explaining the results. Also, indicate the data that is significant different. 

12. It is recommended to avoid using the term functionality in line 370. 

13. The goal of the immunocytochemistry assays was to determine the proliferation capacitary of 3D cell spheroids; however, the authors do not include any quantitative data. It is recommended to compute the spheroid diameter in each time point as an indirect measure of cell proliferation. 

14. Discussion section explains the obtained results and describe a couple of review articles to support them. However, there is lacking a profound discussion of the results. It is recommended to look for similar studies and perform a profound explanation of the results. 

15. The authors evaluated the proliferation capacity of three cell types by varying the stiffness of the NFC hydrogels, reporting differences in cell organization and viability. However, these results may be explained by the fact that all three cell types where under different conditions. As a result, it is not accurate to compare the obtained results. For doing comparisons, it is recommended to evaluate only the effect of stiffness in one type of the cell types used, by employing the NFC hydrogel formations under study. 

16. The manuscript should be checked carefully for grammar. Numerous grammatical errors were identified. Also, some abbreviations are defined several times instead of using them.

17. Continuity in the manuscript could be improved. It is recommended to use more connectors, and finish the paragraphs with a conclusion of the discussion that was described.

Reviewer 3 Report

The authors refer to freeze-dyed gels as aerogels, but aerogels are usually prepared by the supercritical drying method. Gels made by freeze-dyed are cryogels. By another definition, "aerogel" is that the structure of the wet gel is maintained and the dried product is called "aerogel," regardless of the drying method. However, the actual definition of aerogel is unclear as to what degree of a structure must be maintained to be called aerogel. The IUPAC GOLD BOOK defines aerogel as "Gel consisting of a microporous solid in which the dispersed phase is a gas" (https://doi.org/10.1351/goldbook.A00173). According to this definition, aerogels are exclusively porous materials with micropores (<2 nm). However, in reality, a wide variety of materials containing mesopores (2-50 nm) have been reported as "aerogels," and there have been recent papers questioning whether the IUPAC definition is accurate. Some papers question whether the IUPAC definition is not in line with reality (https://doi.org/10.1016/j.micromeso.2017.09.016). At the very least, this paper is a material with a pore of several tens of micrometers, and it is strange to assume that it is an aerogel.

Cryogels usually shrink during the drying process. In this study,  the authors should include photographs before and after drying and after regeneration, which is an important finding in understanding the gel network.

In Fig. 1A, SEM images of the same sample at different magnifications are shown, but it is not useful information for the reader. It would be more useful to that more useful to show SEM images of the same magnification at different NFC concentrations to better understand the gel network.

The storage modulus alone cannot be used to determine if a gel is being formed. In particular, is 0.125% NFC a state that can be considered a gel?

Why do you use gels with different NFC concentrations for different cell types? 

The authors acknowledge that mistakes were made in the automatic segmentation and state that the quantitative evaluation was not possible but was sufficient for a qualitative visual evaluation. However, the authors' initial assertion was that "manual segmentation is subject to the subjective interpretation of the results.“ If a human visually makes the final evaluation, there is room for subjectivity.

The authors state that cryogel has a high shelf life, but how well does it store? In this experiment, how long did you leave the product to refresh after drying?

Round 2

Reviewer 2 Report

I appreciate the response of all the comments. I suggest one more time to revise better the grammatical errors. The writing might be improved. 

Author Response

We thank the reviewer for the feedback and suggestion to revise the grammatical errors of our manuscript once more. We have now revised the manuscript accordingly and fixed the spelling and grammar errors.

Reviewer 3 Report

I think it has been corrected enough and is of good enough quality to be published in polymers.

Author Response

We thank the reviewer for the feedback.